# Initial Tracking, Fast Identification in a Swarm and Combined SLR and GNSS Orbit Determination of the TUBIN Small Satellite

Clément Jonglez *[ID], Julian Bartholomäus[ID], Philipp Werner[ID] and Enrico Stoll[ID]

Chair of Space Technology, Technische Universität Berlin, Straße des 17. Juni 135, 10623 Berlin, Germany
* Correspondence: clement.jonglez@tu-berlin.de; Tel.: +49-30-314-22309

**Abstract:** Flight dynamics is a topic often overlooked by operators of small satellites without propulsion systems, as two-line elements (TLE) are easily accessible and accurate enough for most ground segment needs. However, the advent of cheap and miniaturized global navigation satellite system (GNSS) receivers and laser retroreflectors as well as modern, easy-to-use, open-source software tools have made it easier to accurately determine an orbit or to identify a spacecraft in a swarm, which helps with improving the space situational awareness in orbits that are more and more crowded. In this paper, we present tools for small satellite missions to generate orbit predictions for the launch and early orbit phase (LEOP), identify spacecraft in a swarm after a rideshare launch, and carry out routine orbit determination from multiple sources of tracking data. The TUBIN mission's LEOP phase set a new standard at Technische Universität Berlin: the first global positioning system (GPS) data were downloaded less than four hours after separation, orbit predictions allowed successful tracking by the ground stations, and the spacecraft could be identified in the swarm as soon as the TLE were released by Space-Track. Routine orbit determination from GPS and satellite laser ranging (SLR) tracking data was carried out over several months, and the quality of the orbit predictions was analyzed. The range residuals and prediction errors were found to be larger than those of most SLR missions, which was due to the difficulty of modeling the atmospheric drag of a tumbling, non-spherical spacecraft at low orbital altitudes.

**Keywords:** small satellite; GNSS; SLR; orbit determination; identification; orbit prediction

## 1. Introduction

Since the early 2000s, the number of spacecraft launched on a single rocket has increased from single-digit numbers to over 100 spacecraft with the introduction of dedicated rideshare missions, peaking at 143 spacecraft launched for SpaceX's Transporter-1 mission in 2021. This increase in sheer numbers introduces the difficulty of identifying individual spacecraft in order to contact them in the early days after launch. This is especially relevant for small satellite operators such as start-ups or universities that lack the capability of tracking objects in orbit and that rely on orbit data published by other sources, such as the Combined Space Operations Center (CSpOC).

On 30 June 2021, TUBIN launched in SpaceX's Transporter-2 mission along with 87 other spacecraft. TUBIN is the second installment of the TUBiX20 platform for microsatellites following the launch of TechnoSat as a platform demonstrator on 14 July 2017 [1]. In the instance of Transporter-2, the CSpOC released the first set of two-line elements (TLE) data 8 days after launch, leaving all involved operators having to rely only on the initial state vectors after separation of their own spacecraft's data from orbit. Depending on the capabilities and experience of each individual spacecraft and operating party, an accurate position prediction may thus become difficult or even impossible, and this is followed by the challenge of identifying one's own spacecraft.

Due to an on-board global positioning system (GPS) receiver and the previous experience with the TUBiX20 platform, Technische Universität Berlin (TUB) was able and confident enough in the spacecraft's abilities to commission the GPS receiver system in the second pass over the ground station in Berlin, Germany, and receive the first orbital position data. Consequently, the GPS system was used to generate and download more data over the following days, and it was used to generate our own TLE for use in ground station tracking. Upon the first release of TLE regarding the launch, TUB was able to accurately identify TUBIN based on our own orbit data and propagation methods presented in this paper.

Shortly after launch, the authors started providing orbit predictions to the International Laser Ranging Service (ILRS) network to allow the tracking of TUBIN by satellite laser ranging (SLR) ground stations around the world. The low laser beam divergence of SLR ground stations involves stringent accuracy requirements for orbit predictions. However, because of the different geographic locations, technologies, and level of automation of SLR stations, there is no universal rule on the exact level of orbit prediction accuracy required for a successful station pass [2]. Furthermore, once a given prediction led to successful passes over SLR stations, the time bias service from DiGOS and GFZ Potsdam was able to predict the time bias of this prediction for future passes, which amounts to compensating part of the along-track drift due to atmospheric drag [3].

While dense, geodetic satellites in medium-altitude orbits such as LAGEOS have well-determined orbits with centimeter-level residuals and sub-meter-level prediction accuracy [4], this is not always the case for spacecraft in low-Earth orbits (LEOs). The lower the orbit altitude, the higher the area-to-mass ratio of a satellite, or the higher the solar activity, the faster the orbit prediction can drift because of uncertainty in atmospheric drag modeling, reaching in some cases several kilometers per day [2]. TUBIN, with a relatively high area-to-mass ratio (cross-section from $0.13\,\mathrm{m}^2$ to $0.22\,\mathrm{m}^2$ for a 23 kg mass) and low-altitude orbit (approximately 530 km) and without a controlled attitude most of the time, represents one of most challenging cases for orbit predictions within the ILRS network to the authors' knowledge. Thanks to its GPS receiver, the amount of tracking data is sufficient for ensuring convergence of an orbit estimator, but the issue lies in the rapid degradation of the orbit prediction accuracy.

This paper is divided into five sections. Section 2 introduces the TUBiX20 satellite platform and the TechnoSat and TUBIN missions. Section 3 describes the launch and early orbit phase (LEOP) of the TUBIN mission, including the GPS receiver commissioning, TLE generation from a state vector, first orbit determination from GPS data, and spacecraft identification in a swarm. Section 4 presents the results and lessons learned from several months of GPS- and SLR-based orbit determination that provided prediction data to optical ground stations within the ILRS network. Finally, Section 5 summarizes the findings of this paper and outlines future steps. A list of abbreviations is available at the end of the paper.

## 2. Tubix20 Platform and TUBIN Mission

The TUBiX20 platform aims at providing a modular, fully redundant, and single-failure-tolerant platform for microsatellites in the range from 15 kg to 50 kg. Modularity is implemented in both hardware and software in order to easily adopt the platform for mission-specific requirements. The components and their specific interfacing with the platform's main power and data bus were realized via standardized, redundant TUBiX20 nodes. The removal or addition of platform components thus introduced minimal architectural changes to the platform [5].

### 2.1. Previous Missions of the TUBiX20 Platform

TechnoSat constitutes the first mission based on the TUBiX20 platform. It was launched in July 2017 into a 600 km Sun-synchronous orbit (SSO). As the precursor to TUBIN, it served as a demonstrator for the overall modular platform design and carried only a limited set of attitude determination components as part of the platform. The sensors

included a set of fiber-optic rate sensors, Sun sensors, MEMS gyroscopes, and magnetic field sensors, while the actuation was carried out by four reaction wheels in a tetrahedron configuration, together with three magnetorquers for desaturation. In the absence of a GNSS receiver, TechnoSat had to rely on TLE orbit data uploaded from the ground, and thus the mission faced the challenge of identifying the spacecraft among the 72 small satellites in its launch.

TechnoSat also carried seven technology payloads, most notably a set of 10 mm commercial off-the-shelf (COTS) laser retroreflectors (LRRs) to allow the spacecraft to be tracked via satellite laser ranging (SLR). These LRRs were arranged in different geometrical patterns on six faces of the satellite, enabling the determination of the currently visible faces during laser tracking. The individual reflectors were selected based upon measurements of the reflection characteristics prior to installation. An in-depth analysis of the application of satellite retroreflectors on TechnoSat was carried out in [6].

All reflectors were mounted flat on a total of six faces of the octagonal satellite with a 90° angle between the adjacent faces. This proved to be problematic for laser ranging in cases where the angle of incidence of the laser on a reflector would be larger than 40°, and the return signal would become too weak for most stations to receive. Thus, the reflections from the LRRs left gaps in the tracking of the satellite while its orientation regarding the laser source was changing. As a result, the reflector arrangement on TUBIN was adapted.

A detailed description of TechnoSat and its orbit results can be found in [1].

### 2.2. Goals and Spacecraft Development

The TUBIN mission constitutes the second installment of the TUBiX20 platform. Launched in June 2021, it is tasked with the detection of high-temperature events using microbolometer technology. The payload of TUBIN is formed by a set of imagers sensitive in the visible and thermal infrared ranges of the electromagnetic spectrum. TUBIN is a microsatellite with a launch mass of 23 kg. In order to support its mission, TUBIN was equipped with an improved set of orbit and attitude determination systems. The improved attitude determination and control system includes star trackers and an improved configuration of retroreflectors. TUBIN employs the same actuators as TechnoSat, with the exception of the reaction wheels being set to a higher torque setting. In addition, a GPS receiver is employed within the TUBIN mission: Phoenix, developed by the German Aerospace Center (DLR) [7], which supplies position, velocity, and time (PVT) data in the Earth-centered Earth-fixed (ECEF) WGS84 system.

Both TechnoSat and TUBIN are equipped with several pointing modes, mostly nadir pointing with the camera pointed toward the Earth's center, target pointing to ground stations, or inertial pointing to any target in the True of Date (TOD) coordinate system. Thanks to the modular software architecture, new pointing modes can easily be added to an existing spacecraft by software uploading. When the spacecraft is not used for imaging campaigns or other experiments, its attitude is not controlled (i.e., it tumbles freely).

Position and attitude data are stored within each set of payload data to enable geolocation. The orbit results of TechnoSat demonstrated that the clock error is a large contributor to image geolocation accuracies [8]. Using the GPS receiver as the time provider during payload operations mitigates the clock error and allows for improved accuracy concerning the acquisition of payload data.

Within the TUBIN mission, the retroreflectors of the same type as TechnoSat were rearranged into reflector pyramids that were mounted on both the nadir and zenith side of the spacecraft. These pyramids allowed for an increased field of view and, thus, easier tracking of the spacecraft in orbit. A representation of the retroreflector pyramid assembly is displayed on the right side of Figure 1 below.

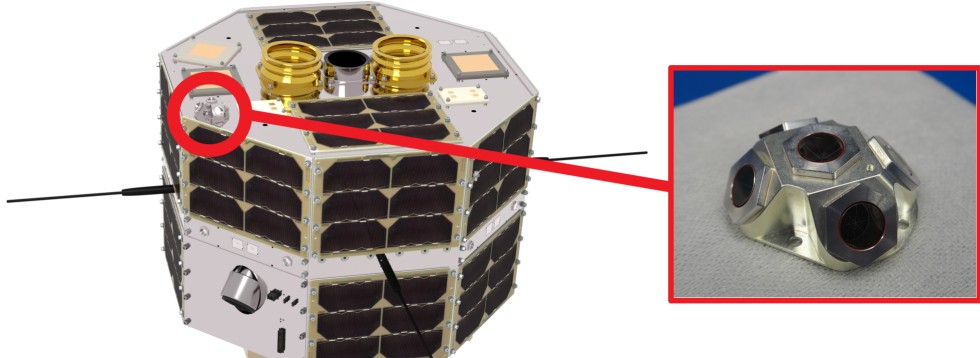

**Figure 1.** (**Left**) Representation of the TUBIN spacecraft in flight configuration. (**Right**) Retroreflector pyramid as it was mounted on nadir and zenith sides.

A detailed description of TUBIN and its initial orbit results can be found in [9]. A summary of the two TUBiX20 missions is shown in Table 1.

**Table 1.** Summary of the TechnoSat and TUBIN satellite missions.

| Mission | TechnoSat | TUBIN |
| --- | --- | --- |
| Objective | Technology demonstration | Technology demonstration<br>Earth observation |
| Initial orbit | 620 km SSO | 530 km SSO |
| Design lifetime | 1 year | 1 year |
| Launch date | 14 July 2017 | 30 June 2021 |
| Spacecraft mass | 20 kg | 23 kg |
| Spacecraft volume | $465 \times 465 \times 305 \text{ mm}^3$ | $465 \times 465 \times 305 \text{ mm}^3$ |
| Orbit determination | Satellite laser ranging (SLR) | SLR<br>GPS receiver |

### 2.3. On-Ground Verification

This section highlights the verification of the laser retroreflectors and of the GPS receiver.

#### 2.3.1. Characterization and Binning of Laser Retroreflectors

For use on TechnoSat and TUBIN, 10 mm-in-diameter LRRs were procured for integration into the spacecraft structure. These retroreflectors are COTS components not specifically manufactured for application in space. The diffraction patterns of the retroreflectors were measured for each unit individually. The most suitable LRRs were selected and integrated into the flight model according to their preferential directions.

#### 2.3.2. GPS Receiver Verification by Spoofing

An end-to-end verification of the GPS receiving chain was carried out by hardware-in-the-loop testing. For this purpose, a passive GPS antenna was placed near the spacecraft and was plugged into a LimeSDR Mini transceiver. The open-source software-defined radio (SDR) simulator LimeGPS was used to generate RF signals mimicking the GPS signals that the spacecraft would receive in orbit [10].

As GPS spoofing is illegal, a Faraday cage was placed around the test area to prevent the simulated GPS signals from escaping to the outside. The transmit power was tuned to match the power of the GPS signals that the spacecraft would receive in orbit.

Not only could the GPS receiver and the cables and antenna be tested this way but the flight software and the ground processing software as well. By comparing the PVT displayed in the telemetry viewer with the PVT used at the input of the GPS SDR

simulator, the coordinate transformations inside the flight and ground software were validated, for example.

The spoofing tests were successful, and the behavior of the GPS receiver proved to be similar in orbit to the simulated GPS signals. For instance, the time to first fix (TTFF) was very close, being slightly more than one minute with a warm start. Here, a warm start means that the receiver was supplied with almanac data and orbit elements corresponding to the simulated orbit and that its clock was configured (within a few seconds of the simulation clock).

## 3. Leop and Spacecraft Identification after Launch

### 3.1. TLE Generation for Ground Station Tracking

Prior to launch, a preliminary state vector containing the PVT of the spacecraft at separation was provided by launch broker Exolaunch. As TUB's ground station network uses TLE for antenna guidance, a conversion from a Cartesian state vector to Brouwer mean elements was necessary. To achieve this, only converting the state vector to Keplerian orbit elements would be wrong, as this would result in osculating elements, whereas the simplified general perturbations SGP4 propagator behind the TLE uses mean elements. Instead, an appropriate approach is the numerical estimation of the Brouwer mean elements. First, the Cartesian state vector is propagated several orbits by a high-accuracy numerical propagator similar to the one described in Section 3.3.1 below. Then, the TLE parameters are tuned using differential correction to match the position and velocity outputs of the numerical propagator as closely as possible.

A TLE generation tool based on this approach and using the Orekit library [11] was written and released as open source [12].

The time window for this propagation and fitting process had to be tuned for better results. The drag coefficient $B^*$ in particular might be very variable for shorter time windows. The optimal time window was found to be 2 days, which is close to the time window used for orbit determination in Section 4 below. With space weather being an important factor in the atmospheric drag for low-Earth orbits, three-hour data from CSSI [13] were used to feed the atmospheric density model. Space weather predictions are more accurate for the near future, and therefore the TLE were generated as shortly as possible before the first expected ground station contact.

The generated TLE were estimated to be reliable enough for several days. The position error between the numerical propagator and its generated TLE is mostly periodic and mostly below 1 km for the first 3 days after the epoch. After 3 days, the error starts to grow larger. Yet, this does not mean that these TLE have an absolute accuracy of 1 km; this only means that it stays within 1 km of the numerical propagator. The error between the prediction and the real satellite's position usually grows larger than that, mostly because of the uncertainties in atmospheric drag.

Less than one hour after spacecraft separation and before the first ground station pass, the post-flight separation vector was received via email. Therefore the preliminary state vector was actually not used, as the post-flight one was from launcher telemetry and was hence more accurate. The TLE generated from the post-flight vector was accurate enough, as it enabled successful ground station contacts for the first 48 h of the mission, until the TUBIN operations team started using GPS orbit determination products instead (cf. Section 3.3 below).

### 3.2. GPS Receiver Commissioning

On the second ground station pass over Berlin, the Phoenix GPS receiver already managed to obtain a fix less than 4 h after spacecraft separation. With the receiver configured for a warm start, the TTFF was only 60 s.

During the first year of the mission, the performance of the GPS receiver was analyzed during several experiments. In nadir pointing (i.e., with the GPS antenna pointing toward the zenith), a warm start always leads to a fix within 90 s. The Phoenix receiver has

12 channels and is therefore able to track up to 12 GPS satellites simultaneously. With a fresh almanac, the receiver managed to track between 10 and 12 GPS satellites most of the time.

Regularly uploading a new set of TLE and almanacs to the spacecraft creates an additional burden for the operations team. Therefore, an experiment was conducted where almanacs of different ages were used for setting up the GPS receiver's warm start. The result was that an almanac up to 3 months old could be used without any loss in tracking performance.

### 3.3. LEOP Orbit Determination from GPS Data

During LEOP, one of the highest priorities was to record enough GPS data in order to perform orbit determination and provide ground stations with orbit predictions while the TLE were not published by CSpOC yet. This is why on the first day of mission, more than 400 GPS PVT data points over several orbits were recorded and downloaded. In the first 10 days of the mission, nearly 700 GPS data points were downloaded in total.

With this GPS data, orbit determination was performed to provide TLE sets to the ground stations. The post-separation state vector was used as a first guess for the batch least squares estimator.

#### 3.3.1. Orbit Determination Model

Table 2 shows a summary of the perturbation forces acting on TUBIN, sorted in decreasing order. This was simulated over a 4 day period with a solar flux of approximately $F_{10.7} = 120$ with TUBIN in nadir pointing for a drag coefficient of 2.2. The solar radiation pressure was modeled here by a cannonball with a single coefficient $C_R = 1.0$. The orbit was a Sun-synchronous orbit at an approximately 530 km altitude. Only accelerations in the along-track direction are shown, as this was the axis that caused issues with the predictions due to atmospheric drag. The underlying models for these perturbation forces are detailed in Table 3 below. Perturbation forces with an acceleration smaller than $10^{-10}\,\mathrm{m\,s^{-2}}$ were not included in this table.

**Table 2.** Overview of perturbation forces on TUBIN, averaged over a 4 day period with $F_{10.7} = 120$, $C_D = 2.2$, and $C_R = 1.0$ in nadir pointing.

| Perturbation Force | Acceleration in Along-Track Direction (Absolute Value, Mean) (m/s²) |
|---|---|
| Earth gravity harmonics 120 × 120 | $7.25 \times 10^{-3}$ |
| Sun third-body attraction | $1.77 \times 10^{-7}$ |
| Moon third-body attraction | $1.68 \times 10^{-7}$ |
| Atmospheric drag | $1.39 \times 10^{-7}$ |
| Solid tides | $5.57 \times 10^{-8}$ |
| Ocean tides | $1.95 \times 10^{-8}$ |
| Sun radiation pressure | $1.37 \times 10^{-8}$ |
| Earth albedo | $3.44 \times 10^{-10}$ |

The Earth's gravity field was the largest perturbation, with the J2 harmonic being the dominant term followed by the Sun and Moon third-body attractions and then closely followed by the atmospheric drag.

However, these values are average values, and the atmospheric drag exhibits high volatility: its maximum value in one orbit is usually five times larger than the minimum value. The atmospheric drag is particularly difficult to model for this mission because the spacecraft has no default attitude and tumbles freely whenever no specific pointing is required (i.e., for downlinks or imaging campaigns). The tumbling attitude amounts to about 97% of an average day. As the satellite is not spherical, this results in unpredictable drag. The attitude is estimated at all times on board the spacecraft and downloaded at regular intervals, so it can be used for orbit determination (which uses data in the past)

but not for future prediction. Attempts were made to predict TUBIN's tumbling behavior, but its residual magnetic dipole is probably the largest source of perturbation torques and is undetermined at the time of writing. As a result, the drag coefficient has to be estimated together with the orbit state vector.

Table 3 shows the parameters and models used in the batch least squares estimator for the orbit determination of TUBIN and other targets.

The orbit determination and propagation scripts written by the authors use the Orekit library that incorporates the models summarized in Table 3. Orekit was used because it is available in Python (via a wrapper from Java), is open-source, and has features and an accuracy rivaling proprietary software, as shown by Ward et al. in 2014 with an earlier version of Orekit [14]. Having access to the source code was an important criterion for the authors to be able to implement the additional features needed for this work. While mostly developed by CS Group, the Orekit project is driven according to an open governance model and welcomes contributions.

**Table 3.** Force models and parameters for TUBIN orbit determination.

| Model or Parameter | Description |
| --- | --- |
| Earth gravity | EIGEN-6S (truncated to 120 × 120) |
| Earth tides | IERS conventions 2010 |
| Ocean tides | FES2004 |
| Third-body attraction | Moon and Sun from DE430 |
| Atmospheric density model | NRLMSISE-00 |
| Drag coefficient | Constant or estimated |
| Space weather data | 3-hourly CSSI data [13] |
| Spacecraft shape | Box-wing model (when attitude available) |
| | Spherical (when no attitude data available) |
| Earth albedo | Knocke model [15] |
| Solar radiation pressure | Lambertian diffusion on each satellite's facet, Equations (8)–(45) in [16] (when attitude available) |
| | Cannonball model, Equations (8)–(44) in [16] (when no attitude data available) |
| Radiation coefficient | Constant or estimated |
| Relativistic corrections | Post-Newtonian (Schwarzschild, Lense-Thirring, de Sitter) [17] |
| Inertial reference system | True of Date |
| Precession and nutation | IAU 2000 |
| Polar motion | C04 IERS |
| GPS data | From TUBIN's Phoenix receiver, quantity and frequency variable |
| GPS antenna—CoG position bias | Applied when attitude data available |
| Numerical integration | Dormand-Prince 853 |
| Integration step size | Variable, max 300 s |
| Orbit determination method | Batch least squares |
| Optimizer | Gauss–Newton with QR decomposer |

Force models can be enabled or disabled as required to incorporate only those models that contribute significantly to the results of the orbit determination. For satellites such as TUBIN, for instance, the prediction inaccuracy is dominated by the atmospheric drag, as outlined in the rest of this work.

The octagonal prism is the defining shape of the TUBIN spacecraft, when neglecting small protrusions such as the UHF antennas or camera baffles.

The drag coefficient can be estimated together with the orbit parameters, but even in this case, it is still constant during the orbit determination window. In the future, methods such as that in [18] will be examined to estimate a time-variable drag coefficient.

As Vallado and Finkleman pointed out [19], atmospheric drag modeling is highly dependent on the input space weather data. Having more frequent input data for the atmo-

spheric density model is especially important for low-altitude orbits, where atmospheric drag is preponderant and highly variable. In order to load three-hourly CSSI space weather data for feeding the atmospheric density model, a new class for the Orekit library was written by the first author, merged into Orekit release 10.2, and is now used by other users. This demonstrates the strength of open-source tools, as anyone can write their own feature and have it merged into a new release if it passes some quality checks. In Figure 2 below, the new CSSI data loader is compared to the legacy MSAFE [20] bulletins, which contain monthly data. This comparison was carried out for a 515 km altitude, Sun-synchronous orbit with an 18 h LTAN. The time window was chosen in October 2014 because the solar activity was high and variable during this month.

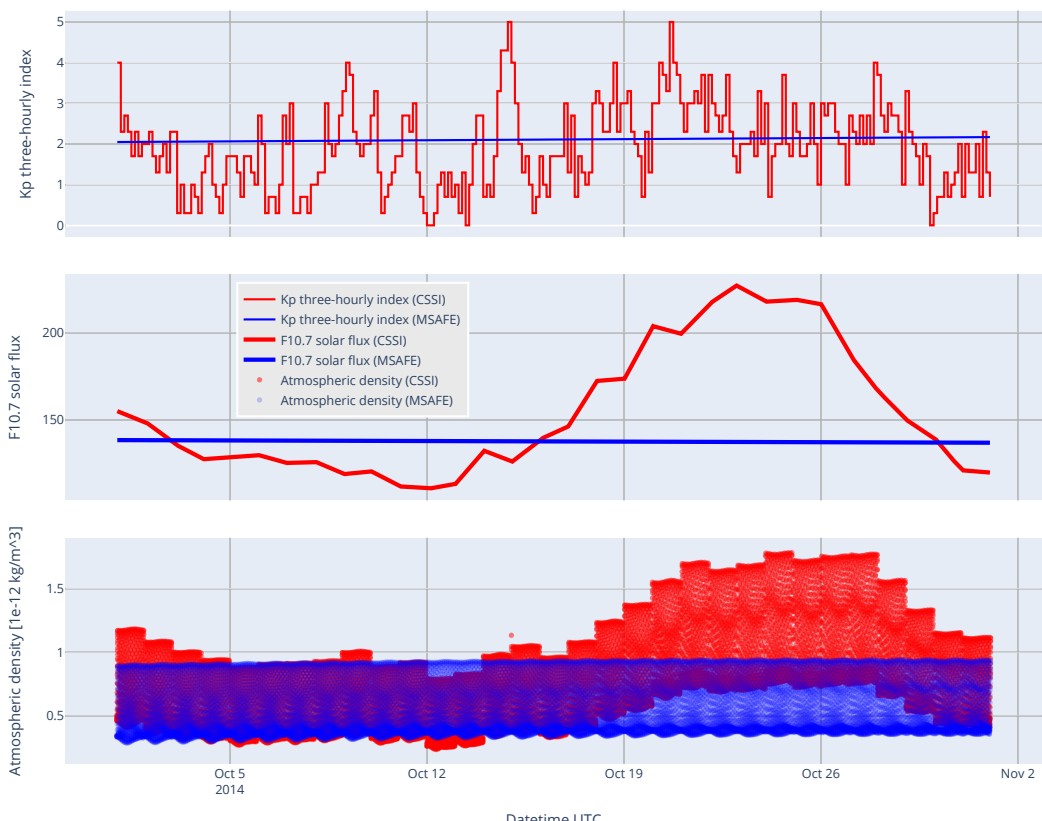

**Figure 2.** Comparison of data and atmospheric density from the CSSI and MSAFE space weather files for an SSO orbit at a 515 km altitude with an 18 h local time of ascending node (LTAN). (**Top**) Three-hour Kp geomagnetic index. (**Middle**) F10.7 daily solar flux (interpolated). (**Bottom**) NRLMSISE-00 atmospheric density model resulting from both data sources.

In Figure 2 above, the Kp and F10.7 data from MSAFE show a linear evolution, which makes sense because the data were interpolated between two monthly entries. The atmospheric density showed a strong correlation with the F10.7 solar flux. The difference between both models was large, at some points being by a factor of more than two, which means that using high-rate data will have an effect on the orbit determination accuracy and very probably a positive effect, although part of the difference can be absorbed by estimating the drag coefficient. The fast variations between the local minima and maxima were due to density variations within one orbit.

Finally, a comparison of atmospheric densities based on the NRLMSISE-00 model was carried out for a variety of Sun-synchronous orbits at different altitudes and LTANs. Due to the near-exponential evolution of the atmospheric density, a rule of thumb was determined following a logarithmic regression: for Sun-synchronous orbits between 400 and 800 km in altitude, a 50 km decrease in altitude results in slightly more than a twofold increase in the

atmospheric drag. This means that TUBIN at a 530 km altitude encountered approximately four times more atmospheric drag than TechnoSat at a 620 km altitude.

### 3.3.2. Verification of the GPS-Based Orbit Determination

Following each orbit determination with new GPS data, TUBIN's orbit was predicted for the following days, and this prediction was saved in CCSDS OEM format in addition to TLE files to avoid the inaccuracies from the SGP4 model. These orbit predictions were compared to subsequent GPS data. For each GPS data point after the prediction epoch, the position error in the LVLH frame (radial, along-track, and cross-track) was computed. Figure 3 only shows the along-track error, which was by far the largest because of the atmospheric drag.

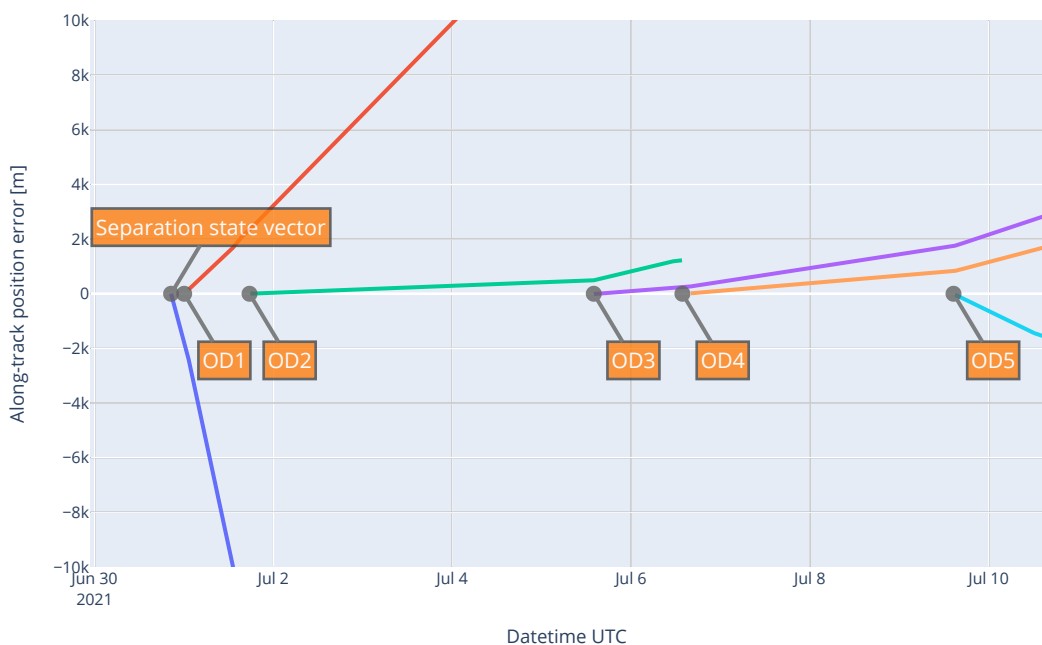

**Figure 3.** Evolution over time of the along-track error between each prediction and subsequent GPS data.

The radial and cross-track errors oscillated within a ±500 m range and a ±50 m range, respectively. As expected for this orbit height (530 km), the along-track error was larger than the radial or cross-track errors and drifted over time.

The prediction computed from only the separation state vector was the worst, as its along-track error increased by 15 km per day. This was because this prediction was based on the propagation of only one state vector, and it did not involve a proper orbit determination. Nevertheless, this prediction still allowed reliable UHF contacts with TUBIN in the first 48 h of the mission.

The first orbit determination (labeled "OD1" in Figure 3 above) was carried out from the first six GPS measurements in a 2 min interval. This limited amount of input data meant that the drag could not be properly estimated, which explains why the resulting prediction showed a significant along-track drift (nearly 4 km per day) but which was nevertheless better than the "state vector"-based prediction.

Figure 4 below shows the estimation residuals of OD2 in the LVLH frame, which are the position difference between the GPS data and the estimated orbit following orbit determination. Low residuals, such as in the order of magnitude of the standard deviation of the input measurements, mean that the least squares estimator converged to a solution that passed "through" the measurements very well. However, this does not necessarily mean that the orbit prediction will be accurate, especially when very few tracking data are available, as was the case, for instance, for OD1. The larger amount of GPS measurements

spread over nearly one day explains why the orbit predictions from OD2 drifted much less in Figure 3 above, with only approximately 200 m of along-track error per day.

One other factor for the relatively low along-track drift of these predictions (particularly OD2 and OD4) is the low solar activity (the observed F10.7 flux was around 90) and, in general, the stable space weather. This resulted in less variation in the atmospheric density and hence in a more predictable drag force.

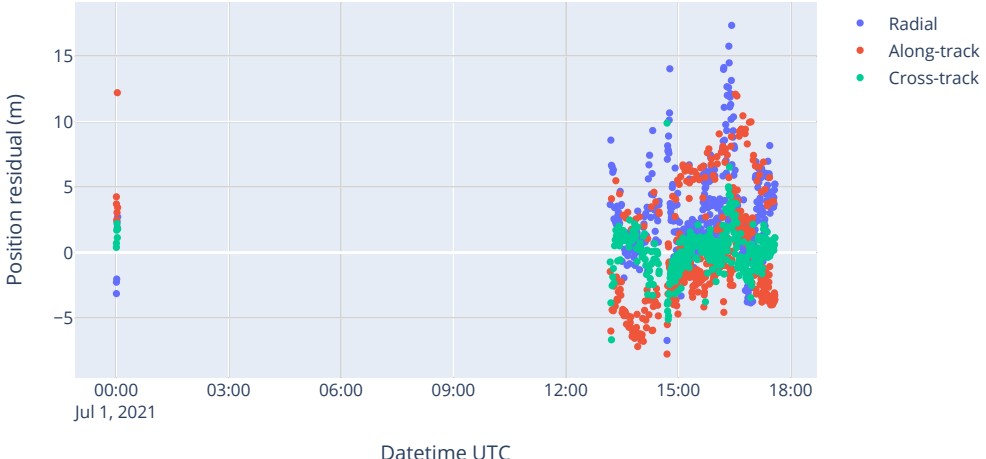

**Figure 4.** Position residuals (GPS data minus estimated orbit) in LVLH frame from the second orbit determination "OD2".

This correlation between the amount of measurement data and the quality of the prediction was also observed for the last three orbit determinations of "OD3", "OD4", and "OD5":

- OD3 was performed using a few tens of GPS measurements over one hour. This explains why this prediction drifted faster than OD2 in Figure 3 above.
- OD4 was performed using continuous GPS measurements from two orbits separated by one day. This orbit determination had the best distribution of measurement data, and therefore, the prediction drifted very little over time.
- OD5 only had 10 min of GPS data at its disposal, which explains why this prediction drifted faster than OD4.

*3.4. Identification in a Swarm*

TUBIN was successfully launched in SpaceX's Transporter-2 mission on 30 June 2021, together with 87 other satellites. In such rideshare missions with a large number of spacecraft, it is difficult for the Combined Space Operations Center (CSpOC) to identify the objects until weeks after launch. For TechnoSat, a previous mission from Technische Universität Berlin without a GNSS receiver, it took 12 days until the spacecraft was identified on Space-Track. During this time, without any other source of tracking data, the TechnoSat operations team had to try out the TLE of different objects from the launch to guide the ground station antenna. If a particular object's TLE led to a poor signal strength or no signal at all at the ground station receiver, then the corresponding satellite was discarded, and the next object in the Space-Track catalog was tried.

Even though Doppler data from ground station passes (over the Technische Universität Berlin's ground station network or even from the SatNOGS open ground station network [21]) could also be used to help identify the spacecraft among the swarm, GNSS data are much more accurate and available in larger quantities because they do not have the constraints of ground station visibility. This involves making sure that the spacecraft is able to record and transmit GNSS data from the first day of the LEOP, which was deemed possible as TUBIN's GPS receiver was extensively tested under simulated orbit conditions, as explained in Section 2.3 above. Additionally, the satellite platform itself and the oper-

ations team gathered a lot of experience and updates to operational procedures, which allowed greater speed in commissioning and operational use of the platform and especially the GPS receiver.

### 3.4.1. Method

To identify which satellite in the swarm was TUBIN, as soon as the TLE for most objects in the launch were published on Space-Track, they were compared to the GPS data downloaded from the spacecraft in the days preceding the TLE. This identification was based on the distance between GPS PVT data and PVT computed from the TLE, with both computed in the same coordinate system. Even if other objects from the launch might be only a few kilometers away from TUBIN shortly after separation, these objects were expected to drift further away, with at least several kilometers per day. Comparing not only the position difference but also the velocity difference makes the method more robust, especially in the first days where the separation between satellites is still small. This method is simple, independent of the orbital element representation or coordinate system, and robust. It is usable even when little GPS data are available, where a batch least squares orbit determination would probably not converge. Furthermore, GPS data were already being recorded for orbit determination purposes, as described in Section 3.3 above, and hence this identification method did not involve any additional overhead in satellite operations, only ground processing.

This approach was successfully tested by the first author in the SALSAT and BEESAT-9 [22] missions from Technische Universität Berlin and was applied again to the TUBIN mission.

### 3.4.2. Results

On 7 July 2021, the first TLE became available for some spacecraft from the Transporter-2 launch, but none of these objects were identified yet. At this point, TUBIN could already be identified by the operations team as NORAD ID 48900 with good confidence. Its distance residuals were below 1 km on average, whereas the second candidate had residuals over 10 km. Most residuals from TUBIN were within 1000 m, which is the usual order of magnitude of accuracy of the TLE.

On 8 July 2021, TLE became available for nearly all objects from the launch, including the three Starlink satellites (NORAD IDs 48879–48881). In addition, seven objects were already identified by their operators. On this day, as the position residuals computed from a new batch of TLE showed NORAD ID 48900 again as the very likely candidate, TU Berlin contacted Space-Track to identify this object as TUBIN.

Figure 5 shows the mean position residuals between each TLE in the launch and GPS measurements from TUBIN on 9 July 2021. As described in Section 3.4.1 above, the velocity residuals were also analyzed, though they are not shown here. A green dot indicates if the corresponding spacecraft was already identified on Space-Track at the time of the plot; otherwise, a red dot is plotted. With the position residuals of the suspected object below 1 km on average, compared with over 50 km for the nearest other object, the identification of TUBIN could be reassured and was finally established on Space-Track on 10 July 2021. This effectively ended the identification efforts as well as this GPS recording campaign, because Space-Track TLE have enough accuracy for Technische Universität Berlin's UHF, S-band, and X-band ground stations.

For other missions without any GNSS receiver but with a ranging-capable transceiver for instance, this method cannot be directly applied. However, it can easily be extended to any type of measurement by generating synthetic measurements (for instance, range) for each object in the same launch and then comparing the synthetic tracking data with the actual tracking data.

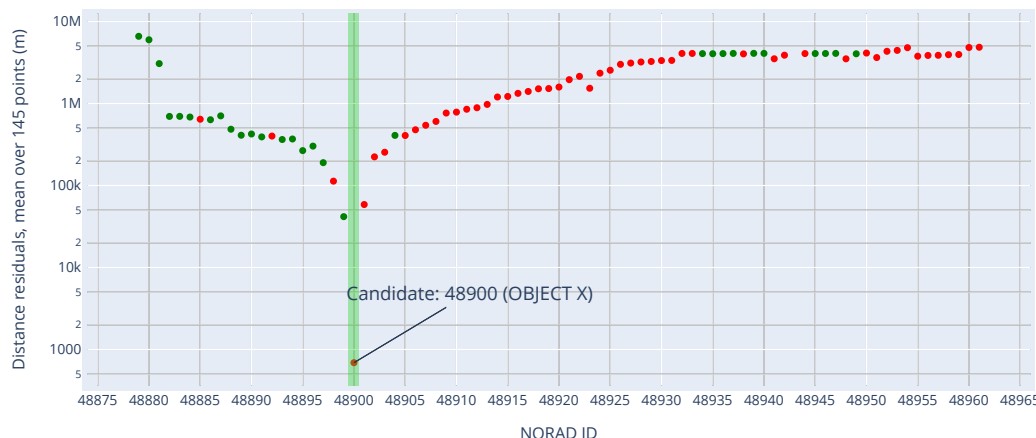

**Figure 5.** Mean position residuals (log scale) between the TLE of each cataloged object in launch and GPS measurements from TUBIN. TLE are from 9 July 2021 at 12:00 a.m. UTC. A green dot means the corresponding spacecraft was already identified on Space-Track at that time, and a red dot means otherwise.

### 3.5. Conclusions on LEOP

TUBIN was identified by its operators on 7 July (i.e., one week after launch) and officially recognized as such by Space-Track 3 days later. However, the operations team would have been able to successfully identify the satellite on the first day of the mission, as GPS data already were recorded during the second ground station pass. Nevertheless, this was not a problem for the LEOP as the TLE generated from GPS-based orbit determination were accurate enough to allow accurate antenna tracking during the first week of mission.

In addition to the GPS receiver working right from the beginning, another factor for the successful LEOP was that the post-launch state vector from SpaceX arrived very quickly (less than one hour after separation), was accurate enough for the first ground station passes, and then served as a first guess for the orbit determination from GPS data.

Finally, quick identification of one spacecraft can help other operators identify their satellites faster by eliminating possible candidates, as Figure 5 above showed with green and red colors.

### 4. Operational Orbit Determination from SLR and GPS Data

The first predictions for TUBIN's orbit were computed based on GPS data only and sent to the ILRS network to get the spacecraft tracked by the SLR stations.

Once SLR data started to become available for TUBIN from October 2021 on, the orbit determination was carried out from both GPS and SLR data. An orbit determination program was written which was able to converge when either no GPS or no SLR data were available in a given time window. Position and velocity data from the TLE were also used to ensure robustness when the measurement data were too sparse. The standard deviation of this TLE feed data was set to a high value, which ensured that the estimator only relied on this data when no GPS and no SLR data were available.

An early version of this program from the SLR data only was released on GitHub with an open-source license and has become the reference tutorial for orbit determination from range data using the Orekit Python wrapper [23].

### 4.1. Models and Parameters for SLR and GPS Orbit Determination

Figure 6 below illustrates the concept of SLR.

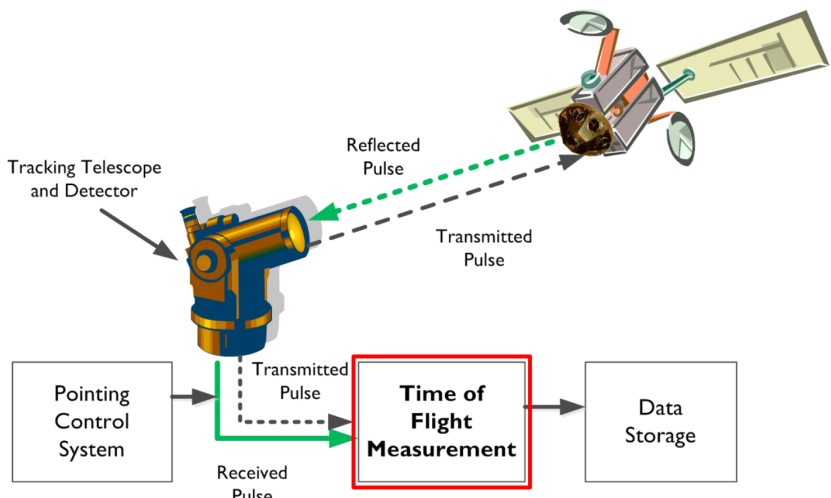

**Figure 6.** Illustration of the concept of satellite laser ranging (SLR) from Kim et al. (2015) [24] (CC BY-NC 3.0).

Table 4 shows the additional parameters and models used for the mixed SLR and GPS orbit determination. Only normal point (NPT) data with a 15 s bin size were used for orbit determination instead of full-rate (FRD) data, because full-rate data proved to increase significantly the computational burden without visibly improving the orbit determination quality. Most works on SLR orbit determination also use NPT data only [24,25].

**Table 4.** Additional parameters for TUBIN orbit determination from SLR range data.

| Model or Parameter | Description |
|---|---|
| SLR station coordinates | SLRF2014 version 200428 + eccentricities |
| SLR data | NPT files, bin size 15 s, from EDC API [26] |
| Tropospheric delay | Mendes–Pavlis model [27] |
| Weather data for tropospheric delay | Included in NPT files |
| Data editing | 7-sigma clipping on SLR data |

The Mendes–Pavlis model [27], available in the Orekit library, was used to model the tropospheric delay of the laser beam, based on weather data published by the SLR ground stations in the CRD pass files.

Classes are available in Orekit to read and write ranging data (CRD format), orbit predictions (CPF format), and SLR station coordinates (SINEX format), following a feature request by the authors. These features were implemented by the Orekit development team, reviewed by the first author and merged into Orekit release 10.3 only a few months after the feature request was opened. This again demonstrates the strength of open-source tools and, in particular, of the Orekit development model.

Data clipping (last row of Table 4 above) was actually not used on TUBIN because the measurement data were too sparse, and this prevented convergence in too many occurrences. For LAGEOS-2 or LARES data, 7-sigma clipping was successful, even though it significantly increased the number of iterations for the estimator to converge.

### 4.2. Verification of SLR-Only Orbit Determination

To verify the accuracy of the orbit determination tool, it was first tested on geodetic satellites such as LAGEOS-2 or LARES. These targets are well-studied and have plenty of SLR data available. Furthermore, their spherical shape makes it easier to model non-gravitational forces such as the atmospheric drag or the solar radiation pressure.

The spacecraft parameters for LAGEOS-2 and LARES such as the mass, cross-section, center of mass offset, and solar radiation pressure coefficient were taken from [25]. A com-

parison of perturbation forces on LAGEOS-1/2, LARES, and other geodetic satellites was carried out in [28].

In the orbit determination example in Figure 7 below, the standard deviation of the range residuals was 30 mm, which is slightly higher than the best analyses available for LAGEOS-2 [25,28]. This could be explained by the lack of empirical accelerations or by the lack of measurement weighting. What is more, the arc length used here was very short (1 day), so the quality of the orbit determination was quite dependent on the distribution of the measurement data [29].

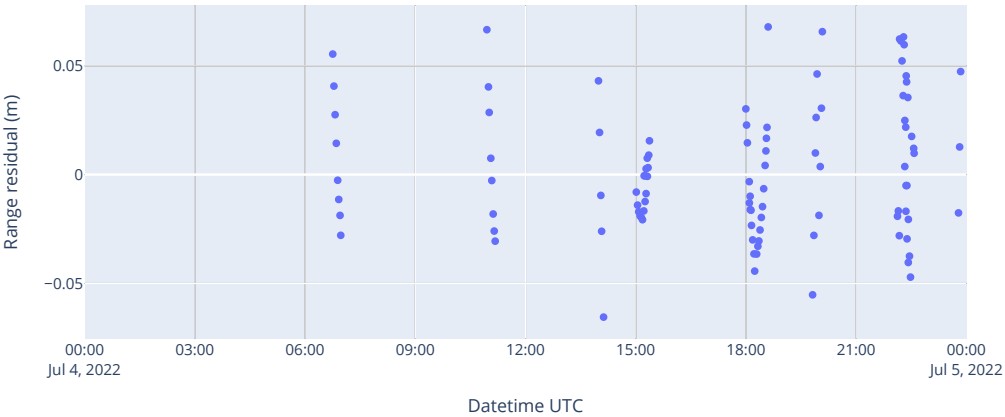

**Figure 7.** SLR range residuals for the LAGEOS-2 spacecraft.

In addition, all orbit predictions generated by the authors' orbit determination tool were automatically compared to the closest CPF prediction used within the ILRS network, if one was available on the CDDIS servers. This consisted of a limited verification of the orbit determination program against other orbit determination software used by prediction centers within the ILRS network, although this verification was less systematic than the cross-validation carried out by Schutz and Tapley on simulated orbit data in [30]. The orbit prediction of LAGEOS-2 corresponding to the example shown in Figure 7 above was compared to a prediction from the same day generated by the Natural Environment Research Council (NERC)'s Space Geodesy Facility (SGF), which is the ILRS prediction center considered to provide the highest quality predictions for LAGEOS-1 and LAGEOS-2 [2]. In this case, the position error after one day compared with SGF's prediction reached approximately 10 cm in the radial direction, 1 m in the along-track direction, and 5 m in the cross-track direction. This was nearly an order of magnitude worse than the SGF's prediction error of LAGEOS-2 shown by Najder and Sośnica in [2]. Nevertheless, no further efforts were attempted to try to improve the prediction accuracy, as the focus of this work was on LEO satellites where atmospheric drag is the dominant source of uncertainty.

*4.3. Quality of the Orbit Determination Products*

After that, the orbit determination program was tested on TUBIN and TechnoSat using SLR data only and without GPS data. The range residuals for TechnoSat and TUBIN were in the order of magnitude of several meters, depending on the amount and the distribution of SLR measurements. This was two orders of magnitude higher than state-of-art SLR orbit determination and was mostly due to the difficulty of modeling the atmospheric drag with a tumbling, non-spherical spacecraft flying at a low altitude, and this will be discussed in depth in this section and further sections below.

This section analyzes the quality of mixed SLR+GPS orbit determination and, in particular, examines the effects of the measurement arc length and of the quantity and weights of tracking data.

To allow for reliable tracking by SLR stations, the time error of a prediction at the time of the ground station pass should be within 10 milliseconds, which corresponds to a 70 m along-track error for LEO. However, in practice, with manual operator tuning or

when using the time bias prediction service from Geoforschungszentrum (GFZ) Potsdam (see Section 4.3.3 below), the acceptable error can be larger. Nevertheless, this accuracy requirement is challenging, especially for a low-altitude orbit with large and variable atmospheric drag.

The covariance of the estimated orbit is given by Orekit and can be propagated. However, we chose to not focus too much on the covariance as it is highly dependent on the standard deviation given for the measurements. In particular, the standard deviation of the SLR measurements is hard to estimate because of the variable and unknown bias between the center of mass and the LRR currently providing the SLR returns, as explained in Section 4.1 above. Instead, the three following subsections describe how the orbit determination products were benchmarked.

### 4.3.1. Residuals

The orbit determination program is able to compute the estimation error covariance matrix. However, this covariance matrix is usually overly optimistic, depending on the weights assigned to the measurement data, as noted by Tapley in Section 4.14 of [31]. This is why this section analyzes the estimation residuals instead. A residual is defined as the difference between an observed (real) measurement and an estimated (synthetic) measurement, whether the measurements are SLR range data or GPS PVT data.

In this section, the effects of the time window length and the amount of tracking data are compared. Figure 8 below shows the range and position residuals, respectively, for a mixed SLR+GPS orbit determination from a 42 h measurement arc.

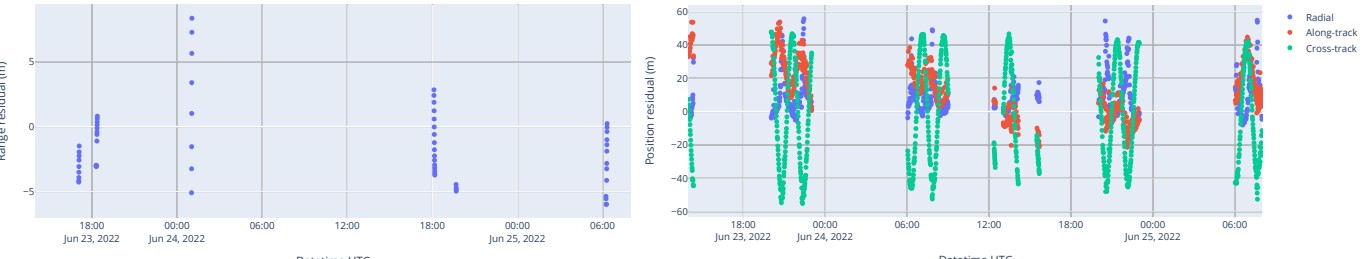

**Figure 8.** SLR range (**left**) and GPS position residuals (**right**) for a mixed SLR+GPS orbit determination on TUBIN on 25 June 2022 from a 42 h measurement arc.

The range residuals in Figure 8 above are to the order of magnitude of several meters, which is high by SLR standards, but this was due to two main factors. First, the atmospheric drag contained large uncertainty, as the TUBIN spacecraft was tumbling most of the time and the orbit determination program did not always have attitude telemetry at its disposal. Second, the relative weights of the GPS and SLR measurements were tuned for robustness rather than accuracy. The estimator had to converge regardless of the amount of measurement data, even if the orbit solution was not the optimal one.

For a shorter measurement time window, such as one day in Figure 9 below, the residuals became lower as the uncertainty of the perturbation forces decreased. This stayed true as long as enough measurement data were available in this reduced time window; otherwise, the estimator would be less likely to converge. In the particular case of Figure 9 below, the three SLR passes were from the same station, which explains the lower residuals. The found solution fit very well with these three SLR passes above the same station but not necessarily with other parts of the orbit.

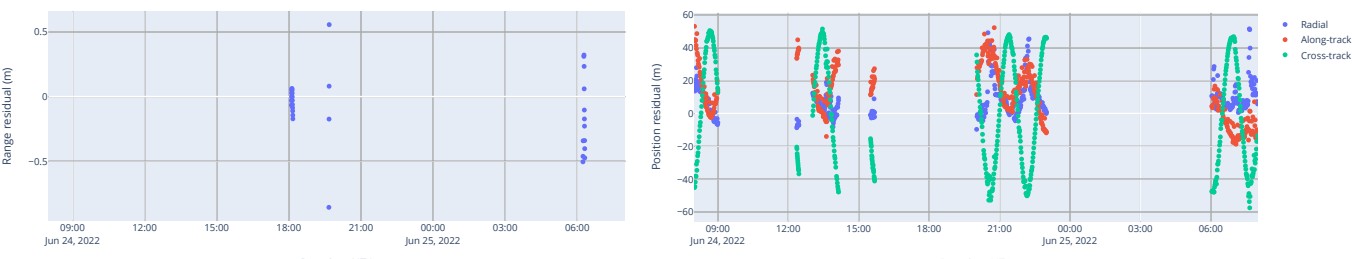

**Figure 9.** SLR range (**left**) and GPS position residuals (**right**) for a mixed SLR+GPS orbit determination on TUBIN on 25 June 2022 from a 24 h measurement arc.

The left side of Figure 10 below shows the position difference between an estimated orbit and the current TLE at the time from Space-Track. As expected for such a low-altitude orbit, the error was the largest in the along-track direction because of the atmospheric drag. However, even the radial and cross-track errors were to the order of several hundreds of meters. This amount of cross-track error in particular could be problematic for an SLR ground station, as the pointing requirements are stringent. This poor accuracy explains why TLE are usually not used by SLR ground stations to track spacecrafts, except for a first acquisition when no other source of prediction is available but with low chances of success per pass.

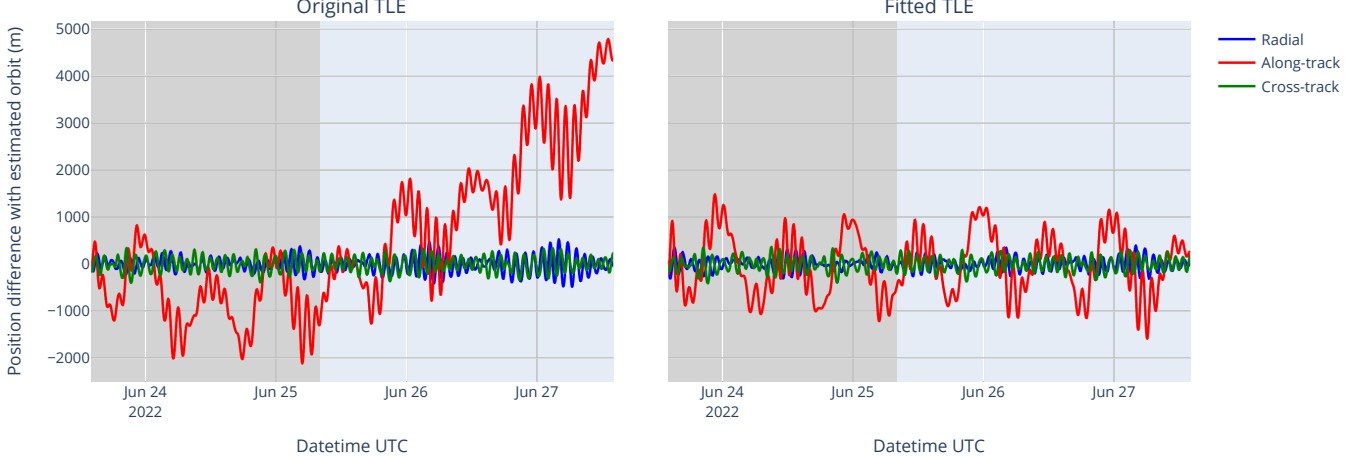

**Figure 10.** Position difference between an estimated orbit and two sets of TLE for an orbit determination on 25 June 2022 with a 42 h arc. The darker gray area on the left of each figure represents the orbit determination window. (**Left**) Original TLE from Space-Track. (**Right**) TLE optimized by differential correction. The along-track error of the optimized TLE remained mostly periodic and within a $\pm 1$ km range in the time interval represented.

The orbit determination program is able to generate CPF and CCSDS OEM files for a precise tracking of the spacecraft by ground stations. However, for some legacy ground station systems that are only able to use TLE, it can be interesting to generate "improved" TLE based on the estimated orbit. Therefore, the orbit determination program performed the same operation as described in Section 3.1 above to fit a set of TLE as well as possible to the estimated orbit.

The right side of Figure 10 above shows the position difference between the estimated orbit and the "improved" TLE. The along-track error remained mostly periodic and within a $\pm 1$ km range. This corresponded approximately to the accuracy limit of the SGP4 model, which was mostly due to the neglection of the tesseral m-daily terms [32].

### 4.3.2. Comparison of Successive CPF Predictions

The orbit determination program was automatized and ran every day for several months in the first half of 2022. After the estimator converged, the software generated Consolidated Prediction Files (CPF, both version 1 and 2), which it uploaded to the CDDIS servers to allow the SLR ground stations to accurately track TUBIN.

For consistency, each new orbit determination was compared to the previous CPF prediction.

For instance, Figure 11 shows the position difference in LVLH axes between two consecutive CPF predictions on 24 and 25 June 2022. The along-track difference grew by approximately one kilometer per day in this case. Even though this position difference is not an absolute error, it is a good indicator of the order of magnitude of the prediction quality.

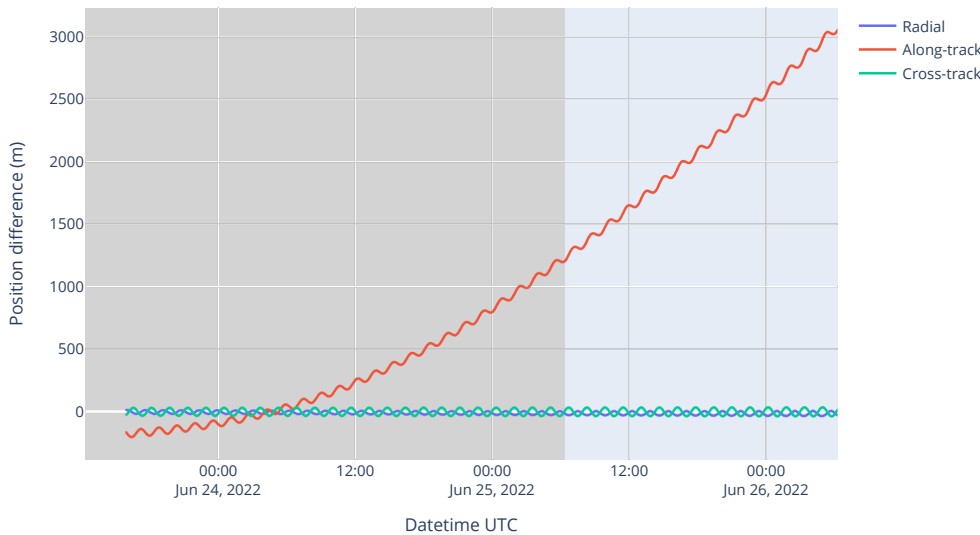

**Figure 11.** Position difference between the estimated orbit and the previous CPF prediction for an orbit determination on 25 June 2022 with a 42 h arc. The darker gray area on the left represents the orbit determination window.

The variability between consecutive CPFs was, most of the time, a few kilometers, which corresponds to several hundreds of milliseconds of along-track time error. This means that when a new CPF is published to the ILRS network, the predicted position of TUBIN often "jumps" by a few kilometers, even though it is not a real jump as SLR ground stations do not switch CPFs during a pass.

This large variability is another demonstration of the difficulty in modeling the atmospheric drag with an unknown future spacecraft attitude. Other estimators such as the sequential batch least squares method (available in the Orekit library since release 11.0) should reduce these jumps, and they will will be investigated in future studies.

### 4.3.3. Time Bias of Orbit Predictions

DiGOS and GFZ Potsdam developed a time bias service for all satellites tracked by the ILRS network [3]. The time bias of an orbit prediction is defined by the along-track error, which is the largest error for low-altitude orbits, divided by the orbit velocity. Each CPF prediction for each target was compared to the SLR data from each ground station pass, using this CPF prediction to determine the time bias of this prediction at the time of the pass. This process is similar to the analysis of the GPS prediction along-track error shown in Figure 3 above, except that the range data did not allow a direct position comparison. Hence, an extra step was required here. The time bias was determined by a least squares method, which tuned the time bias to make the actual range data fit the predicted range data as close as possible. Therefore, this method actually equals to computing the residuals of measurements acquired after the orbit determination was performed, except the range

residuals were not directly analyzed but used to estimate the along-track error at the time of the pass.

Once at least two passes are available for a given CPF, a trend can be computed via polynomial fit to predict the time bias for future passes [3]. This time bias trend, represented as a red line in Figure 12 below, was then used by SLR stations in the following passes to have an improved orbit prediction of the target. In this figure, the GPS residuals (converted to a time error in the along-track direction) are also shown for comparison to the SLR-based extrapolation, and both matched pretty closely, which shows that the polynomial trend method from [3] worked pretty well.

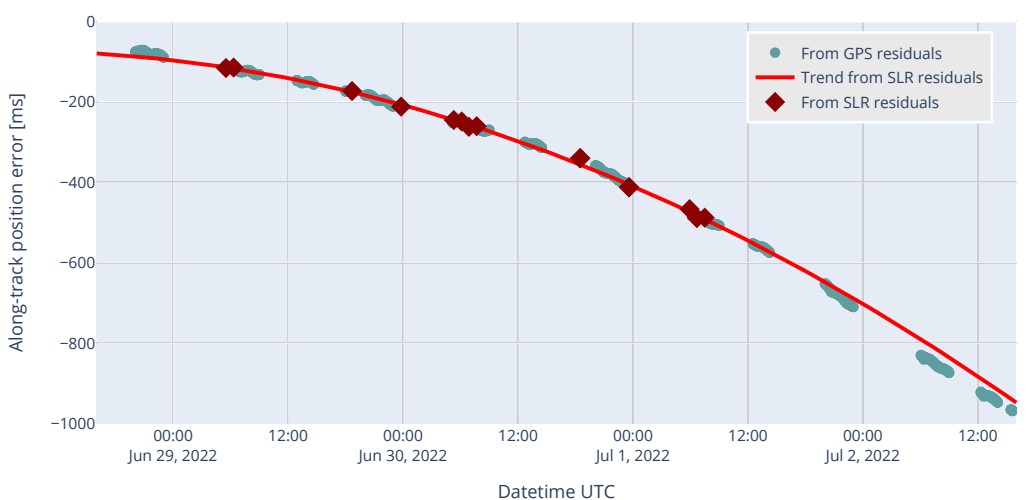

**Figure 12.** Evolution of time bias of TUB17901 CPF orbit prediction generated on 28 June 2022. Teal represents time bias computed from GPS measurements, dark red represents time bias from SLR data (from GFZ website), and the red line represents the polynomial fit from SLR time bias to predict the time bias trend (from GFZ website).

Figure 12 also shows that TUBIN could still be tracked, whereas the orbit prediction had several hundreds of milliseconds of time bias. Normally, above a few tens of milliseconds of time bias, it became much harder for the SLR station to find the satellite. In this case, as the time bias was known by the SLR station thanks to the time bias prediction service from GFZ Potsdam, the satellite could still be found by the SLR station.

This time bias information is a very useful reality check and can be used to improve past predictions. It is indeed possible to use real SLR data to tune past predictions and make them match future measurement data more closely, such as by tuning the drag model. This technique is similar to setting a zero weight to part of the tracking data, which are then not used by the estimator but are used to give an indicator of the quality of the orbit determination. The time bias service was, for instance, used with success to improve the quality of TechnoSat predictions at the beginning of 2018 [3]. Since it has been implemented within the ILRS network, this time bias service helped increase the rate of successful passes for all missions and therefore the data output of the ILRS network.

4.3.4. Improvement of Orbit Prediction Using GPS Time Bias

A measurement campaign was carried out in the first two weeks of September 2022, where the spacecraft was placed in nadir pointing and GPS data were recorded nearly all the time. These GPS data, converted to time bias, were used to analyze the drift of orbit predictions. A significant advantage over SLR-based time bias is that SLR data are only available from ground station passes.

Having near-continuous GPS data allowed tuning the orbit determination program, particularly the estimation window, as each different prediction was compared to subsequent GPS data. The effect of the drag coefficient was also analyzed. Figure 13 below shows

the evolution of the orbit prediction error for different values of the drag coefficient (this coefficient is normally estimated together with the orbit parameters).

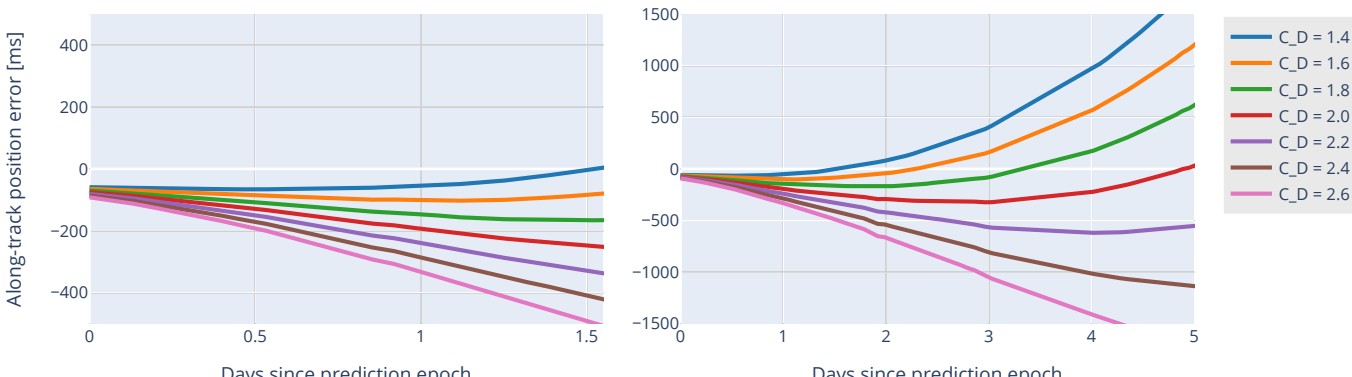

**Figure 13.** Evolution of time bias of multiple predictions with different drag coefficients $C_D$ for TUBIN on 6 September 2022, measured by comparison to subsequent GPS data. (**Left**) First 36 h after the prediction epoch. (**Right**) First 5 days after the prediction epoch.

Some predictions provide better results for a short period after the epoch, such as with $C_D = 1.4$ in Figure 13 on the left, whereas other predictions will prove to be better only after a few days, such as with $C_D = 1.8$ or $C_D = 2.0$ in Figure 13 on the right. This shows the limits of the cannonball drag model. For better results, the drag coefficient should be variable, as performed in [18], for instance.

### 4.4. SLR Data Statistics

In the first 9 months of SLR tracking of the TUBIN mission, nearly 3000 NPT data points were gathered by the ILRS network, as shown in Figure 14 on the right below.

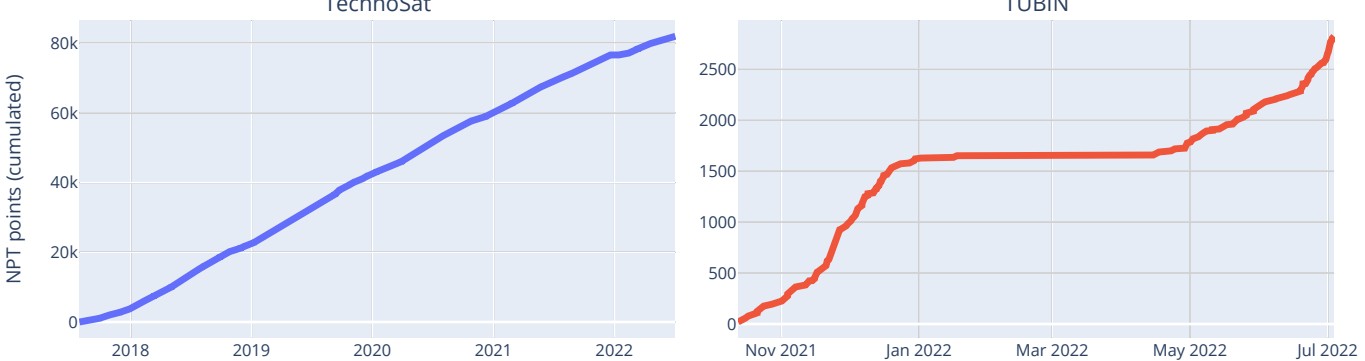

**Figure 14.** Cumulated number of normal points since the beginning of mission until 4 July 2022 for TechnoSat (**left**) and TUBIN (**right**).

TUBIN was lost by the ILRS network between January and April 2022 because the SLR data became too sparse to allow robust orbit determination. The lack of SLR data was in turn due to the orbit predictions having a decreasing accuracy and then not being possible at all anymore. To overcome this chicken-and-egg problem, and to start tracking TUBIN again, Technische Universität Berlin published orbit predictions based on GPS data to the ILRS network. As enough SLR tracking data became available, DLR started publishing SLR-based orbit predictions again.

Unlike TUBIN, the TechnoSat mission was tracked nearly without interruption in the 5 years of the mission, as the linear shape of the cumulative function of the NPT points shows in Figure 14 on the left above. This could be due to TechnoSat's orbit resulting in approximately four times less atmospheric drag than TUBIN's, as estimated in Section 3.3.1 above, which caused a slower degradation in prediction accuracy from the drag uncertainty.

*4.5. Effect of the Attitude on the Atmospheric Drag*

To confirm the suspicion that the large time error of the predictions was indeed due to the unmodeled spacecraft attitude and therefore a large atmospheric drag uncertainty, a simulation was written to compute the time difference between two hypothetical spacecrafts. These two satellites started from the same initial orbit, which was TUBIN's Sun-synchronous orbit at a 530 km altitude. The spacecrafts' pointing laws were aligned with the local orbit frame LVLH, but each had a different angular offset to ensure that the first pointed its side of the minimum cross-section toward the velocity direction, whereas the second pointed its side toward the maximum cross-section. These two extreme cases of atmospheric drag allowed us to give an upper limit of the along-track deviation caused by the attitude.

The same numerical propagator, together with a box model of the TUBIN spacecraft as described in Table 3 above, was used. The minimum and maximum cross-sections were $0.13\,\mathrm{m}^2$ and $0.22\,\mathrm{m}^2$, respectively, which meant a ratio of approximately 1.6 between both values.

The epoch chosen was on 15 June 2022, which had average solar activity with an observed F10.7 flux around 140.

Figure 15 below confirms that the predictions' quality quickly degraded due to the unmodeled attitude. The along-track time bias could reach nearly 200 ms after one day and reached over 600 ms after two days, which in most cases prevented the SLR stations from finding the spacecraft. This evolution was approximately square with the time since the epoch. In reality, this drift might be even faster, as the atmospheric density in LEO can have fast and strong local variations which are not covered by the atmospheric model of this simulation.

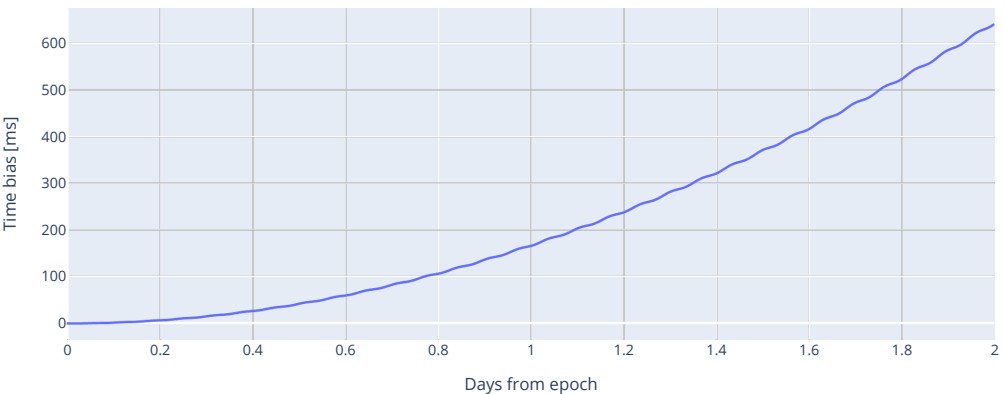

**Figure 15.** Simulated along-track drift between two initially identical orbits but with different pointing laws, converted to a time difference in milliseconds, as a function of the number of days since epoch. The orbit was TUBIN's Sun-synchronous orbit at a 530 km altitude. The spacecrafts were pointing nadir with each having a different offset that ensured the minimum and maximum cross-sections, respectively, with regard to atmospheric drag.

## 5. Conclusions and Future Work

In this work, we presented results from the first year of operations for the TUBIN mission, with a particular focus on flight dynamics tasks such as spacecraft identification and orbit prediction.

The challenges of predicting the orbit of a high-drag tumbling spacecraft were explained. In the future, other orbit determination methods such as sequential batch least squares will be tested in the hope to improve the accuracy of orbit predictions.

Another key improvement could be to estimate a time-variable drag coefficient, as was performed in [18] for instance. A variable drag coefficient based on Fourier series, for instance, would allow us to compensate, at least partially, for the drag uncertainty due to the spacecraft tumbling. Although this method requires attitude information for the

best results, it can still be applied based on past data to potentially capture frequencies associated with periodic variation of the drag coefficient in orbit due to repeating attitude and ambient parameters. This knowledge of periodic variations could then be used to predict the future drag coefficient even when the spacecraft is tumbling.

Finally, these analyses were carried out using only open-source software, which shows the potential of open-source solutions for precision flight dynamics.

Flight dynamics experience gathered in the TUBIN mission will be useful for future missions of Technische Universität Berlin, such as the QUEEN mission, which will be equipped with a multi-constellation GNSS receiver.

**Author Contributions:** Conceptualization, C.J., J.B. and P.W.; methodology, C.J.; software, C.J.; formal analysis, C.J.; investigation, C.J.; resources, C.J. and P.W.; data curation, C.J.; writing—original draft preparation, C.J., J.B. and P.W.; writing—review and editing, C.J., E.S., J.B. and P.W.; visualization, C.J. All authors have read and agreed to the published version of the manuscript.

**Funding:** The TechnoSat and the TUBIN missions were funded by the Federal Ministry for Economic Affairs and Energy (BMWi) through the German Aerospace Center (DLR) on the basis of a decision of the German Bundestag (Grant No. 50 RM 1219 and 50 RM 1102).

**Institutional Review Board Statement:** Not applicable.

**Informed Consent Statement:** Not applicable.

**Data Availability Statement:** Input SLR data are available from the open access EUROLAS Data Center (EDC) [26].

**Conflicts of Interest:** The authors declare no conflict of interest.

## Abbreviations

The following abbreviations are used in this manuscript:

| | |
|---|---|
| CDDIS | Crustal Dynamics Data Information System |
| CoG | Center of gravity |
| CPF | Consolidated Prediction Format |
| CRD | Consolidated Laser Ranging Data Format |
| CSpOC | Combined Space Operations Center |
| DLR | Deutsches Zentrum für Luft- und Raumfahrt |
| ECEF | Earth-centered Earth-fixed |
| EDC | EUROLAS Data Center |
| FOV | Field Of view |
| GFZ | GeoForschungsZentrum |
| GNSS | Global navigation satellite system |
| GPS | Global positioning system |
| ILRS | International Laser Ranging Service |
| LEO | Low-Earth orbit |
| LEOP | Launch and early orbit phase |
| LRR | Laser retroreflector |
| LVLH | Local-vertical local-horizontal |
| MEMS | Microelectromechanical systems |
| MSAFE | MSFC Solar Activity Future Estimation |
| MSFC | Marshall Space Flight Center |
| NERC | Natural Environment Research Council |
| NORAD | North American Aerospace Defense Command |
| NPT | Normal point data |
| POD | Precise orbit determination |
| PVT | Position velocity and time |
| QUEEN | QUantentechnologien für den Einsatz auf Einem Nanosatelliten |
| SDR | Software-defined radio |
| SGF | Space Geodesy Facility |

| SGP4 | Simplified general perturbations |
| SINEX | Solution Independent Exchange |
| SLR | Satellite laser ranging |
| TLE | Two-line elements |
| TOD | True of date |
| TTFF | Time to first fix |
| TUBIN | TU Berlin Infrared Nanosatellite |
| UHF | Ultra-high frequency |

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
