# Peer review of "Initial Tracking, Fast Identification in a Swarm and Combined SLR and GNSS Orbit Determination of the TUBIN Small Satellite"

_aerospace, doi:10.3390/aerospace9120793_

Round 1
Reviewer 1 Report
The manuscript is solid, very well written and presents real flight experience. I definitely support its publication following some minor changes and clarifications.
Apart from technical remarks provided below, I am a bit concerned with complete lack of the literature review. Although not necessary for the present material, it may benefit the article. It’s up to authors to decide whether to include some review or not.
1. What is the attitude of the satellite? Is it completely random and uncontrolled, or at least passive magnetic system is installed? Nadir pointing is mentioned few times, but how it is achieved? Some information appears on Lines 203-208, but it is too late and incomplete.
2. Line 197: Table 2 provides Earth gravity influence for 120 terms in the series, while text states only J2. Obviously, J2 is the dominant term, but technically this contradicts Table 2.
3. Line 213: Orekit appears out of nowhere. Probably better to introduce it with something like “the orbit propagation is performed with Orekit software that incorporates models summarized in Table 3”. Also, is Orekit developed by the manuscript authors, or did they only modify/add some models/libraries for this tool? Not clear from the text (lines 224-228 might be expanded with this regard).
Okay, line 400 clarifies this. However, I keep the comment above as it is, since the question arises a lot earlier.
Overall, Orekit and its implementation in the work requires better introduction.
4. What does “applied” mean for the solar radiation pressure in Table 3, while other factors have model names?
5. Lines 347-355 essentially repeat lines 341-346.
6. Line 575: The uncertainty is probably the same: the fraction of the error in the total value. But the value is less, so the value of the error is less as well.
7. Typos at lines 69, 488, 523.
Author Response
Thank you for the review.
A short literature review was added to the introduction.
- TechnoSat’s and TUBIN’s actuators are now described in section 2.1 and section 2.2, respectively. A description of the pointing modes was added in section 2.2.
- The formulation was changed to clarify this.
- This should now be fixed, more content was added about Orekit’s development model and about the authors’ contributions to Orekit.
- A reference to the equation used by Orekit’s implementation of solar radiation pressure was added.
- This is now fixed.
- The formulation was changed to clarify this.
- This is now fixed.
Reviewer 2 Report
This paper presented the results of the orbit determination for the TUBIN mission. Although the method and algorithm used here is not new, the author showed us how they did in the operation, and the process and the results is clear.
Author Response
Thank you for the review.
Round 2
Reviewer 1 Report
I have no additional remarks.
Author Response
Thank you.